# Multi-PQTable for Approximate Nearest-Neighbor Search

**Xinpan Yuan [1], Qunfeng Liu [2], Jun Long [2],*, Lei Hu [2] and Songlin Wang [1]**

[1]    School of Computer, Hunan University of Technology, Zhuzhou 412000, China; xpyuan@hut.edu.cn (X.Y.);
       WangSL110@163.com (S.W.)
[2]    School of Computer Science and Engineering, Central South University, Changsha 410083, China;
       qunfengliu@csu.edu.cn (Q.L.); hudalei@csu.edu.cn (L.H.)
*    Correspondence: jlong@csu.edu.cn; Tel.: +86-0731-8253-9926

**Abstract:** Image retrieval or content-based image retrieval (CBIR) can be transformed into the calculation of the distance between image feature vectors. The closer the vectors are, the higher the image similarity will be. In the image retrieval system for large-scale dataset, the approximate nearest-neighbor (ANN) search can quickly obtain the top k images closest to the query image, which is the Top-k problem in the field of information retrieval. With the traditional ANN algorithms, such as KD-Tree, R-Tree, and M-Tree, when the dimension of the image feature vector increases, the computing time will increase exponentially due to the curse of dimensionality. In order to reduce the calculation time and improve the efficiency of image retrieval, we propose an ANN search algorithm based on the Product Quantization Table (PQTable). After quantizing and compressing the image feature vectors by the product quantization algorithm, we can construct the image index structure of the PQTable, which speeds up image retrieval. We also propose a multi-PQTable query strategy for ANN search. Besides, we generate several nearest-neighbor vectors for each sub-compressed vector of the query vector to reduce the failure rate and improve the recall in image retrieval. Through theoretical analysis and experimental verification, it is proved that the multi-PQTable query strategy and the generation of several nearest-neighbor vectors are greatly correct and efficient.

**Keywords:** image retrieval; ANN; product quantization; mutil-PQTable

## 1. Introduction

With the rapid development of mobile internet and social multimedia, images and videos are growing explosively every day. How to quickly and accurately obtain similar images in large-scale image dataset has become a hot and difficult topic for multimedia researchers. Image retrieval or content-based image retrieval (CBIR) can be transformed into calculating the distance of their feature vectors. The closer the feature vectors are, the higher the similarity of the image is. In traditional machine learning, the main methods of extracting image features are Scale-Invariant Feature Transform (SIFT) [1,2], Speeded Up Robust Features (SURF) [3,4], GIST descriptors [5,6], Fisher Vector [7], Vector of Locally Aggregated Descriptors (VLAD) [8,9]. In deep learning, the main methods include Convolutional Neural Network (CNN) [10,11], Siamese Network [12–14], Triplet Network [15–17]. However, the image feature vectors extracted by these methods are all high-dimensional. It is difficult to satisfy the performance requirement directly through brute search or linear scanning. In order to reduce the computing time, the approximate nearest-neighbor (ANN) search becomes feasible.

In the image retrieval system for large-scale image dataset, when the dimension of the image feature vector increases, the computing time will increase exponentially due to the curse of dimensionality. The traditional ANN algorithms, such as KD-Tree [18], R-Tree [19], and M-Tree [20], perform poorly

when dealing with high-dimensional image feature vectors. Their performance is not even as good as that of linear search [21]. The locally sensitive hashing (LSH) algorithm solves the problem of high-dimensional vector search from another angle. It encodes high-dimensional vectors into a fixed-length hash code through a series of hashing functions and calculates the similarity between images quickly by hamming distance [22,23], while it does not make full use of the data itself during the construction of hashing functions and the generation of hashing codes. If the hashing algorithm wants to obtain a high retrieval accuracy, the length of the hash code needs to be long enough. This will reduce the collision probability of similar samples during random transformation and reduce the recall rate.

Recently, Herve Jegou et al. proposed the product quantization (PQ) algorithm for nearest neighbor search [24]. PQ is a popular and successful method to compress a high-dimensional vector into a short code (e.g., 32 bit). Meanwhile, PQ can quickly calculate the approximate distance between the original vector and the compressed code by symmetric distance computation (SDC) or asymmetric distance computation (ADC). Under the same compression ratio, the retrieval accuracy of the PQ algorithm is higher than that of the hashing algorithm. After the PQ algorithm was proposed, a series of related quantization algorithms were developed to improve the computing speed and accuracy for ANN search, such as, the Optimized Product Quantization (OPQ) [25,26], Locally Optimized Product Quantization (LOPQ) [27], Stacked Quantization (SQ) [28,29], and Additive Quantization (AQ) [30] algorithms. Besides, some research work has been performed about how to parallelize the image retrieving process [31,32].

Inspired by the successes of the PQ and its extensions for ANN search, our main contributions are as follows:

1.  We propose a product quantization table (PQTable) algorithm on the basis of the PQ algorithm, according to the ability of the Hash Table to quickly find the required content. This algorithm can implement a non-exhaustive approximate nearest-neighbor search algorithm, aiming at quickly and accurately retrieving the vector candidate sets in a large-scale dataset.
2.  We also propose a multi-PQTable query strategy for ANN search. Besides, we generate several nearest-neighbor vectors for each sub-compressed vector of the query vector to reduce the failure rate and improve the recall in image retrieval.

The rest of the paper is organized as follows: Section 2 discusses the product quantization algorithm. Section 3 describes the multi-PQTable in detail. Section 4 verifies the correctness and efficiency of the algorithm through experiments. Section 5 gives conclusions.

## 2. Product Quantization

In 2011, Herve Jegou et al. proposed a PQ algorithm based on vector quantization (VQ). The PQ algorithm speeds up image retrieval in large-scale image datasets. In the PQ algorithm, the product refers to the cartesian product, and the quantization refers to vector quantization.

Given a set $X$ and a set $Y$, their cartesian product is also a set composed of all the ordered pairs from the set $X$ and the set $Y$, which can be recorded as $X \times Y$:

$$X \times Y = \{(x, y) | x \in X \wedge y \in Y\} \tag{1}$$

Let $X = [x_1, x_2, \ldots, x_D]$, $X \in R^D$, where $X$ represents a D-dimension vector. The quantization process for the vector $X$ can be expressed as:

$$q(X) \in C = \{c_i | i = 1, 2, 3, \ldots, k\} \tag{2}$$

where $q(.)$ denotes a quantization function, the set $C$ represents a codebook of length $k$, and the element $c_i$ is a codeword or centroid.

In the process of vector quantization, the quantization error $e(x)$ is usually expressed by the minimum mean-square error, as shown in Equation (3). The smaller the mean-square error, the better the quantization performance.

$$e(x) = min\|q(x) - x\|^2 \tag{3}$$

The PQ algorithm decomposes a high-dimensional vector into several low-dimensional vectors and constructs their Cartesian product. Then, the PQ algorithm quantizes and compresses these low-dimensional vectors separately through the *K*-means algorithm. The product quantization process for the vector *X* is shown as in Equation (4).

$$\underbrace{x_1, \ldots, x_{D^*}}_{u_1(x)}, \ldots, \underbrace{x_{D-D^*+1}, \ldots, x_D}_{u_M(x)} \tag{4}$$
$$q_1(u_1(x)), \ldots, q_M(u_M(x))$$

The process in detail is as follows:

(1)  Uniformly split the vector *X* into *M* distinct sub-vectors $u_j(x)$, $1 \le j \le M$. The dimension of the sub-vector is *D\** and *D\** = *D/M*, where *D* is a multiple of *M*. Therefore, the vector *X* can be seen as a series of sub-vectors, and $X = [u_1(x), u_2(x), \ldots u_M(x)]$.

(2)  Each sub-vector is quantized and compressed by the *K*-means algorithm, and the corresponding codebook set $C_j$ is obtained.

(3)  The Codebook *C* of the vector *X* is the Cartesian product generated from all the set $C_j$, and $C = C_1 \times C_2 \times \cdots \times C_M$.

After quantizing the vector, product quantization provides two methods for quickly calculating the distance of vectors: SDC and ADC, as shown in Figure 1.

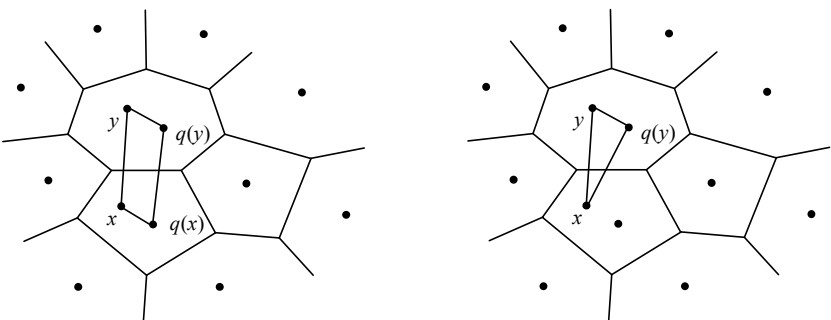

**Figure 1.** Symmetric distance computation and asymmetric distance computation.

**SDC**: the vector *x* and the vector *y* can be represented by their corresponding centroids $q(x)$ and $q(y)$. The distance $D(x, y)$ between vector *x* and vector *y* can be approximated to the distance $D(q(x), q(y))$ between their centroids $q(x)$ and $q(y)$. That is to say, $D(x, y) \approx D(q(x), q(y))$, as shown in Equation (5).

$$D(x, y) \approx D(q(x), q(y)) = \sqrt{\sum_j^M D\big(q_j(x), q_j(y)\big)^2} \tag{5}$$

where $D(x, y)$ denotes the Euclidean distance between the vector *x* and the vector *y*. $D(q_j(x), q_j(y))^2$ can be quickly obtained from the lookup table according to the index value of the *j*-th sub-quantizer. The lookup table contains the square of the distance between all the sub-quantizer centroids.

**ADC**: this method only needs to represent the vector y with its centroid $q(y)$. The distance $D(x, y)$ between the vector x and the vector y can be approximated to the distance $D(x, q(y))$. That is, $D(x, y) \approx D(x, q(y))$, as shown in Equation (6).

$$D(x, y) \approx D(\mathrm{x}, q(\mathrm{y})) = \sqrt{\sum_{j}^{M} D\left(u_j(x), q_j\left(u_j(y)\right)\right)^2} \tag{6}$$

## 3. Multi-PQTable for ANN Search

This section detailly introduces the multi-PQTable for ANN search. Section 3.1 briefly describes how to convert image retrieval into distance calculation between feature vectors. Section 3.2 records the PQTable algorithm, mainly including the process of constructing the PQTable and vector search. Section 3.3 describes the multi-PQTable query strategy in detail.

### 3.1. Problem Description

Given a query image $I_q$ and the image dataset $I = \{I_1, I_2, \ldots, I_N\}$, $N$ represents the size of the dataset.

The target of the ANN search is to quickly obtain the top-$k$ images, namely, the candidate dataset $Y = \{Y_1, Y_2, \ldots, Y_k\}$, which are closest to the query image $I_q$ from the dataset $I$. The image retrieval task can be transformed into vector retrieval, as shown in Figure 2. The specific steps of image retrieval are as follows:

(1)  Extract the features for the query image $I_q$ and for the image dataset $I$ by feature extraction tools such as SIFT, GIST, CNN, and so on. Correspondingly obtain the image feature vector $Q = [q_1, q_2, \ldots, q_n]$ and the image feature dataset $X = \{X_1, X_2, \ldots, X_N\}$, where Q is the feature vector of the query image $I_q$, $X_i = [x_{i1}, x_{i2}, \ldots, x_{iD}]$, and $D$ is the dimension of the feature vector.

(2)  Obtain the top-$k$ vector candidate subsets $S_c = \{S_1, S_2, \ldots, S_k\}$ through calculating and sorting according to the query vector $Q$.

(3)  Correspondingly obtain the top-$k$ image candidate sub-dataset $Y = \{Y_1, Y_2, \ldots, Y_k\}$ via the linking relationship between the vectors and the images.

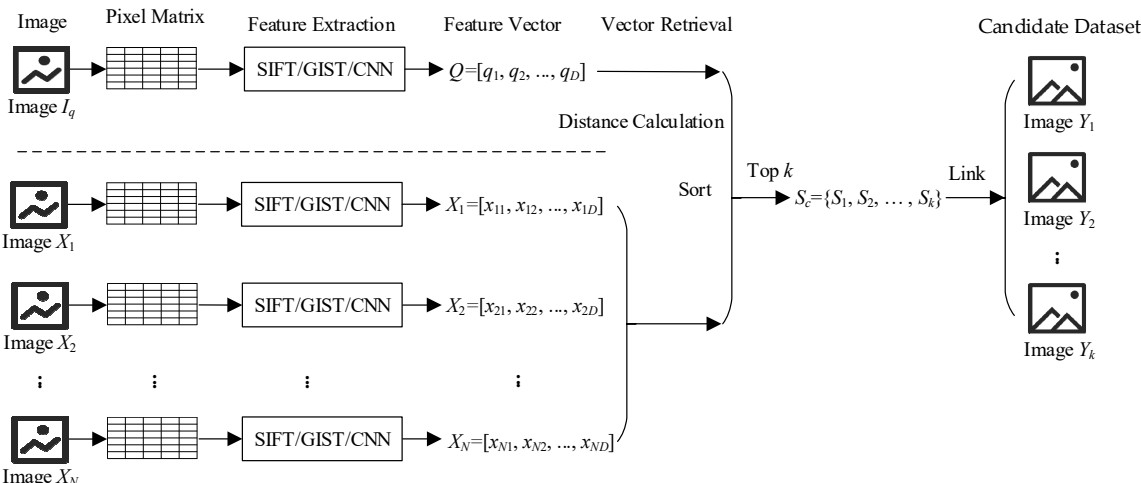

**Figure 2.** The image retrieval task is transformed into vector retrieval.

According to the PQ algorithm, the computational cost of SDC and ADC is O($KD + NM$), which is fast for small $N$ and still linear in $N$. However, when $N$ is very large, the calculation time is still long. In order to further reduce the computation time and improve the retrieval efficiency, this paper proposes a product PQTable algorithm on the basis of PQ algorithm, according to the ability of the Hash Table to quickly find the required content. This algorithm can implement a non-exhaustive

approximate nearest-neighbor search algorithm, aiming at quickly and accurately retrieving the vector candidate sets in massive databases.

### 3.2. PQTable Algorithm

The main idea of the PQTable algorithm is to construct the vector index structure of the PQTable, after quantizing and compressing the vector by the PQ algorithm. The PQTable algorithm can quickly obtain the candidate sets through a look-up table. In the process of vector quantization and compression, it is assumed that the number of clustering centers of the *K*-means algorithm is *K*.

Given a vector dataset $X = \{X_1, X_2, \dots, X_N\}$, $X_i \in R^D$, $X_i = \left[X_i^1, X_i^2, \dots, X_i^M\right]$. $PQ(X_i)$ represents the product quantization process of vector $X_i$, as shown in Equation (7).

$$X_i \rightarrow PQ(X_i) = \left[PQ\left(X_i^1\right), PQ\left(X_i^2\right), \dots, PQ\left(X_i^M\right)\right] \tag{7}$$

where $X_i \in R^D$, $X_i^j \in R^{D/M}$, $1 \le i \le N$, $1 \le j \le M$.

Each sub-vector $X_i^j$ has *K* choices during quantization and compression. After the set *X* is quantized and compressed by the product quantization algorithm, we can create an $L \times M$ two-dimensional product quantization table (PQTable), as shown in Figure 3, where $L = K^M$.

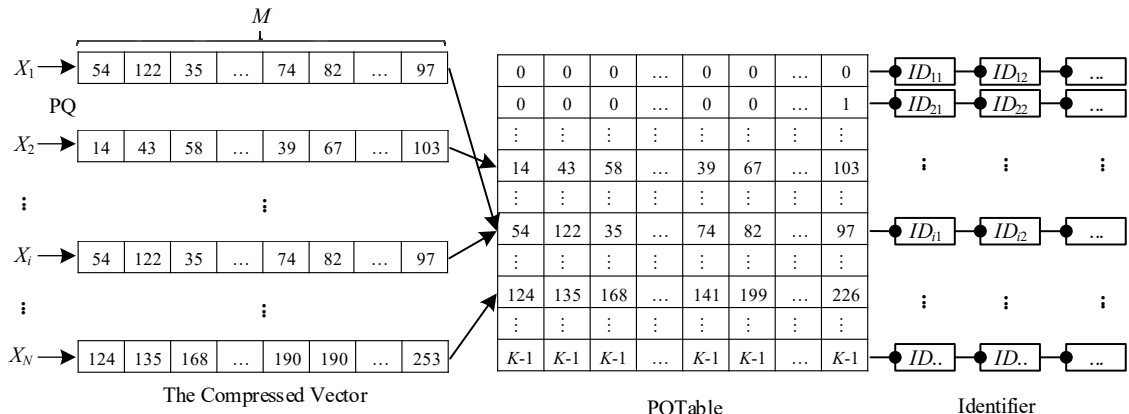

**Figure 3.** The process of constructing the product quantization table (PQTable).

The vector retrieval process in the PQTable is shown in Figure 4. The steps of vector retrieval in detail are as follows: Firstly, each sub-vector of the query vector *Q* is quantized by the quantizers $q = \{q_1, q_2, \dots, q_m\}$, and we can obtain a compression vector $PQ(Q)$, where the quantizers come from the quantization process of set *X*. Then, we can use the compression vector $PQ(Q)$ to search in the PQTable. Finally, we can quickly get the candidate set according to the mapping relationship between the original vector and the compressed vector.

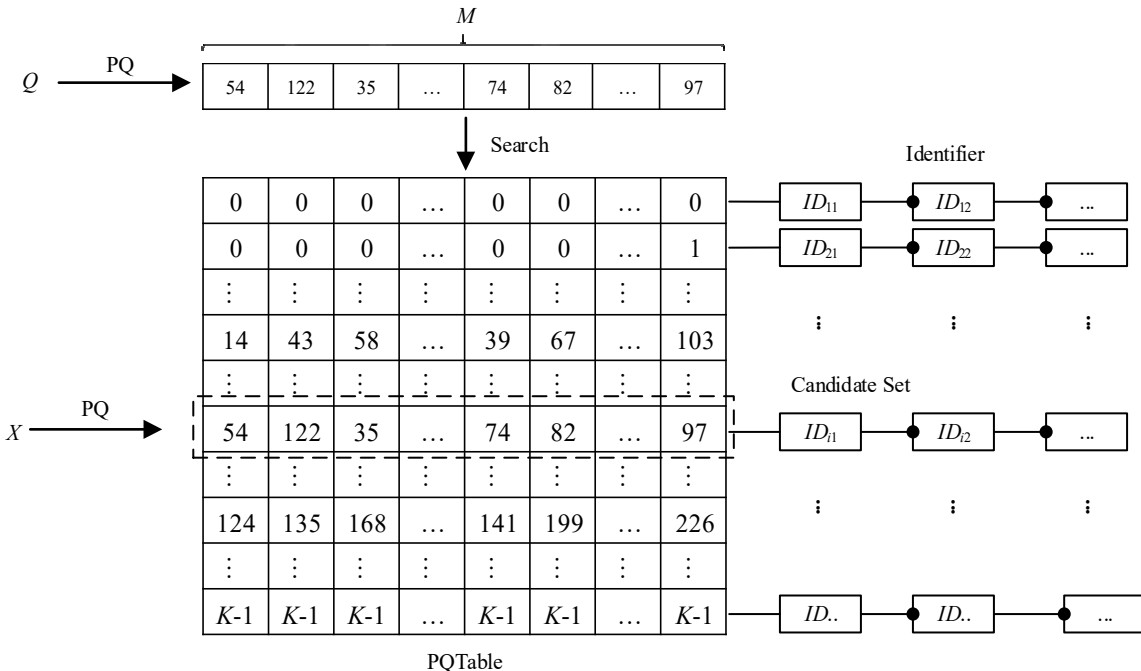

**Figure 4.** The process of vector retrieval in the PQTable.

### 3.3. Multi-PQTable Query Strategy

The PQTable is a two-dimensional table with the size of $L \times M$, where $L = K^M$. The computational cost of retrieval vectors in PQTable is O($LM$), and the computational cost of the PQ algorithm is O($KD + NM$) ≈ O($NM$). When $N = 1 \times 10^9$, $K = 256 = 2^8$, $M = 8$, $L = K^M = (2^8)^M = 256^M = 256^8 = 1.84 \times 10^{19}$, there is obviously $L >> N$. Thus, their computing time is sorted as $O(LM) >> O(NM)$. In other words, the PQTable increases the computational time of vector retrieval. When $L >> N$, the identifiers in the PQTable are mostly empty and sparse, which easily leads to retrieval failure.

In order to solve these problems, this article proposes a PQTable query strategy and generates several nearest-neighbor vectors.

1. The Multi-PQTable Query Strategy

We divide the compressed vector into T sub-compressed vectors whose size is $M^* = M/T$. Then, we construct a sub-quantized table for each sub-compressed vector. In this way, we can get $T$ sub-PQTables, and the size of the table is $L^* \times M^*$, where $L^* = K^{M^*}$. When searching via the query vector Q, the compression vector $PQ(Q)$ is also divided into $T$ sub-compression vectors, namely, $PQ(Q) = [subPQ(Q)^1, subPQ(Q)^2, \ldots, subPQ(Q)^T]$. Then, we query the $subPQ(Q)^t$ in the $subPQTable^t$ and get the corresponding candidate set $S_C^t$, where $1 \leq t \leq T$. Finally, we union the $T$ candidate sets and obtain the final candidate set $S_C$, as shown in Equation (8).

$$S_C = S_C^1 \cup S_C^2 \cup \cdots \cup S_C^t \cup \cdots \cup S_C^T \tag{8}$$

When $T = 2$, $M^* = M/T = M/2$, there is $L^* = K^{M^*} = (2^8)^{M/2} = (2^4)^M = 16^M = 16^8 = 4.2 \times 10^9$. When $T = 4$, $M^* = M/T = M/4$, there is $L^* = K^{M^*} = (2^8)^{M/4} = (2^2)^M = 4^M = 4^8 = 65,536$. Apparently, $L^* << L$, and $O(L^*M) << O(LM)$. According to the above analysis, it is found that the multi-PQTable query strategy greatly reduces the computational time and effectively improves the speed of vector retrieval. When $T = 2$, the process of the PQTable query strategy is as shown in Figure 5.

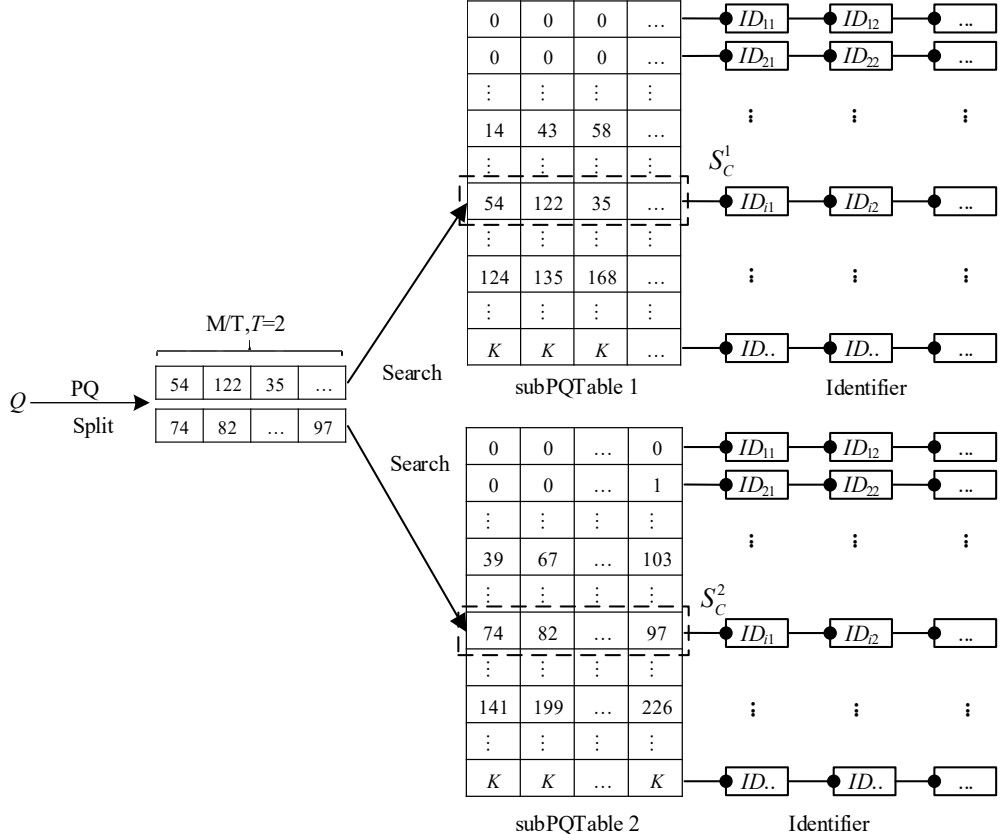

**Figure 5.** Process of the PQTable query strategy when $T = 2$.

2. Generating Several Nearest-Neighbor Vectors

Based on the multi-PQTable query strategy, this paper generates several nearest-neighbor vectors (*NNV*) for each sub-compressed vector $subPQ(Q)^t$ of the query vector $Q$. The purpose is to reduce the failure rate and improve the recall rate. The specific steps for generating several neighbor vectors are as follows:

(1) Creating a generator table. We define $K^* - 1$ as the largest parameter and create a two-dimensional generator table with a size of $U \times V$, where $K$ is a positive integer, and $K^* << K$, $U = K^{*M^*}$, $V = M^*$, $M^* = M/T$. We sequentially fill the table with integers from 0 to $K^* - 1$ and get a generator table (GenTable).

(2) Generating several nearest-neighbor vectors. Each $subPQ(Q)^t$ is added and subtracted to every row data in the GenTable. Then, we can get the corresponding nearest-neighbor vector set (NNVS).

(3) Filtering the elements in nearest-neighbor vectors. We validate each vector in the *NNVS* and filter out the vectors whose elements are less than zero. Finally, we get the final *NNVS*.

When $K^* = 4$, $M^* = 4$, $subPQ(Q)^t = [54,122,35,63]$, $U = 4^4 = 64$, $V = 4$; the process of generating *NNVS* is shown in Figure 6. The generating nearest-neighbor vector set (genNNVS) algorithm is shown in Algorithm 1.

**Algorithm 1.** The input of the algorithm includes the generator table *GenTable*[$U$][$V$] and the sub-compression vector $subPQ[V]$. The output is the nearest-neighbor vector set *NNVS*. Lines 3–10 shows that each $subPQ(Q)^t$ is added and subtracted to every row data in the GenTable. Combining lines 7–9 with lines 12–14, we obtain that the algorithm filters out the vectors whose elements are less than zero.

---

**Algorithm 1**: Generating Nearest-Neighbor Vector Set (genNNVS).

---

Input:
       *GenTable*[*U*][*V*], *subPQ*[*V*]

Output:
       *NNVS*

1:    for *i* <= *U* do
2:       Flag = true;
3:       for *j* <= *V* do
4:          $V_0[j]$ = *subPQ* [*j*] + *GenTable*[*i*][ *j*];
5:          $V_1[j]$ = *subPQ* [*j*] − *GenTable*[*i*][*j*];
6:          if $V_1[j]$ < 0 then
7:            flag = false;
8:          end if
9:       end for
10:      *NNVS* add $V_0$;
11:      if flag == true then
12:        *NNVS* add $V_1$;
13:      end if
14:    end for

---

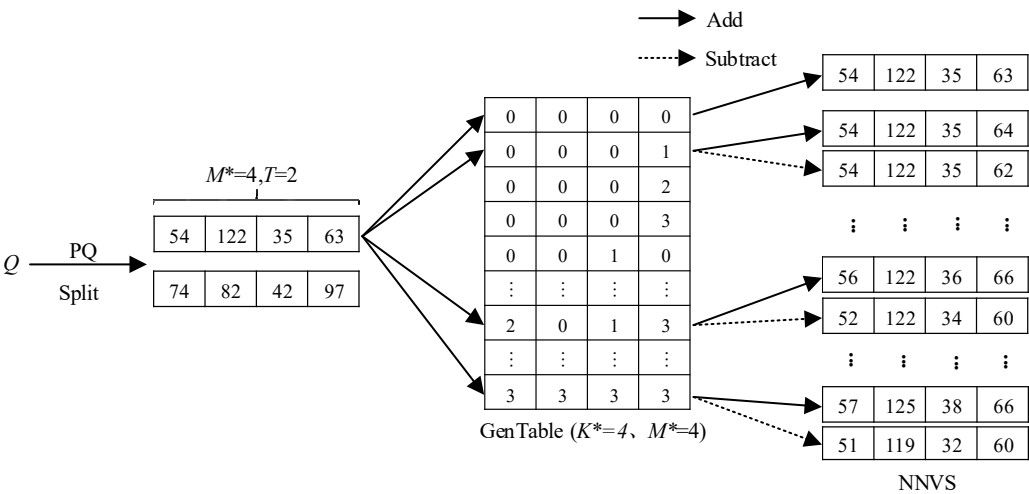

**Figure 6.** Process of generating *NNVS* when $K^* = 4$, $M^* = 4$, $subPQ(Q)^t$ = [54,122,35,63].

3.    Implementation of the Multi-PQTable Query Strategy

     Given the query vector *Q* and the vector dataset $X = \{X_1, X_2, \dots, X_N\}$, the process of quickly obtaining the candidate set $S_C$ through the multi-PQTable is shown in Figure 7. Besides, the approximate nearest-neighbor search algorithm based on the multi-PQTable is shown in Algorithm 2.

     **Algorithm 2.** The input of the algorithm includes the candidate set size *k*, the query vector *Q*, the generator table *GenTable*[*U*][*V*], and *T* *subPQTables*, where *subPQTables* = {*subPQTable*[1], *subPQTable*[2], $\dots$, *subPQTable*[T]}. The output is the candidate set $S_{Ck}$, whose size is *k*. Line 1 indicates that the query vector q is quantized and compressed by the PQ algorithm. Line 2 indicates that the compressed vector PQ(*Q*) is split into T sub-compressed vectors, i.e., *subPQs* = {*subPQ*[1][*V*], *subPQ*[2][*V*], $\dots$, *subPQ*[T][*V*]}. Line 5 represents the generating NNVS for each sub-compressed vector *subPQ*[t][*V*]. Lines 6–10 describe that the algorithm searches all *vector* in the *subPQTable*[t] and gets the corresponding candidate set $S_C^t$, where $1 \le t \le T$. Line 13 indicates that the algorithm obtains the final candidate set $S_{Ck}$ through calculating similarity and sorting for their original vectors. The algorithm can also set the similarity threshold (e.g., α = 0.5) in advance and only add the vector whose similarity is greater than the threshold to the candidate set $S_{Ck}$.

---

**Algorithm 2**: The Approximate Nearest-Neighbor (ANN) Search Algorithm based on the Multi-PQTable

---

Input:

    *k*, *Q*, *GenTable*[*U*][*V*], *subPQTables* = {*subPQTable*$^1$, *subPQTable*$^2$, ... , *subPQTable*$^T$}

Output:

    $S_{Ck}$

1:   *PQ*(*Q*);

2:   *subPQs* = {*subPQ*$^1$[*V*], *subPQ*$^2$[*V*], ... , *subPQ*$^T$[*V*]};

3:   while *t* <= *T* do

4:      *t* = *t* + 1;

5:      $NNVS^t$ = genNNVS(*GenTable*[*U*][*V*], *subPQ*$^t$[*V*]);

6:      foreach *vector* ∈ *NNVS* do

7:         *IDs*←search *vector* in *subPQTables*$^t$;

8:         *sc*←obtain the vector set through *IDs*;

9:         $S_C{}^t = S_C{}^t \cup sc$;

10:     end foreach

11:     $S_C = S_C \cup S_C{}^t$;

12:  end while

13:  $S_{Ck} = S_C$;

14:  size←get size of $S_C$;

15:    if size > *k* then

16:    $S_{Ck}$←calculate similarity and sort;

17:  end if

---

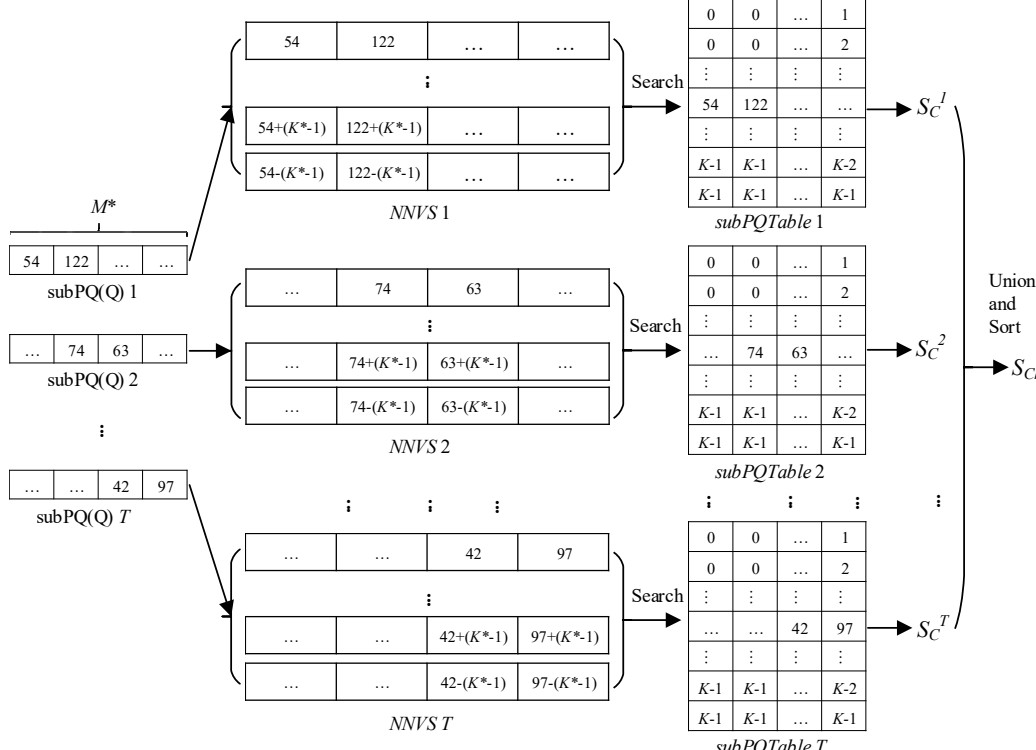

**Figure 7.** The process of obtaining the candidate set SC through the multi-PQTable.

## 4. Experiments and Analysis

This section verifies the correctness and efficiency of the algorithm model through experiments. Section 4.1 briefly describes the experimental settings, including the experimental dataset and the network model for extracting image features. Section 4.2 introduces in detail the experimental results of the ANN search algorithm based on the multi-PQTable.

### 4.1. Experimental Settings

In our previous research work, we proposed the Triplet Spatial Pyramid Pooling Network (TSPP-Net) through combing the triplet convolution neural network with the spatial pyramid pooling [33], which can process any size images without cutting or scaling, as shown in Figure 8. The network model improves the generalization ability of the network and the accuracy of the image similarity measurement.

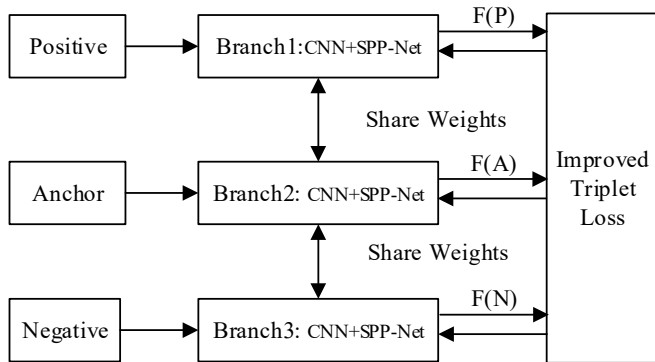

**Figure 8.** The construction of the Triplet Spatial Pyramid Pooling Network (TSPP-Net).

Besides, we improved the original learning target and proposed a new learning goal, as shown in Figure 9. The new goal can achieve twice the distance learning, including minimizing the distance between an anchor and a positive, maximizing the distance between an anchor and a negative, and maximizing the distance between a positive and a negative.

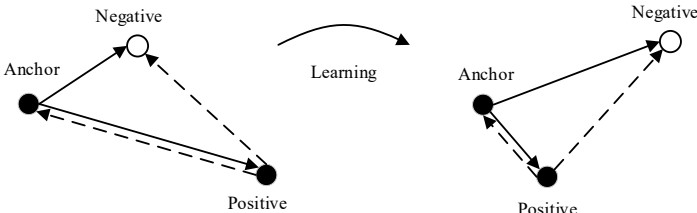

**Figure 9.** The improved learning goal of the Triple Network.

On the basis of the original triple loss function and the new distance learning goals, we easily developed the improved triplet loss function, as shown in Equation (9). Compared with the original triple loss function, the improved triple loss function can realize twice the distance learning only through a triple sample.

$$
\begin{aligned}
L\left(x^a, x^P, x^n, \alpha, \beta\right) \quad &= \frac{1}{N}\sum_{i}^{N} max\left\{D\left(x_i^a, x_i^p\right) - D\left(x_i^a, x_i^n\right) + \alpha, 0\right\} \\
&+ \frac{1}{N}\sum_{i}^{N} max\left\{D\left(x_i^p, x_i^a\right) - D\left(x_i^p, x_i^n\right) + \beta, 0\right\}
\end{aligned}
\tag{9}
$$

In this paper, we also use the TSPP-Net model to extract image features from Caltech 101 dataset and obtain the corresponding image feature vectors. We analyze the effect and performance of the multi-PQTable algorithm in the approximate nearest-neighbor search of the image. However, the Caltech 101 dataset only contains 9146 images, which approximately corresponds to $1 \times 10^4$ levels. Therefore, we randomly select an image from the MNIST dataset and obtain a one-dimensional pixel matrix. We add some random factors into the one-dimensional pixel matrix and repeat them three times to synthesize a three-dimensional pixel matrix. We input the three-dimensional pixel matrix into the TSPP-Net model and get the corresponding feature vectors. After repeating the process for

$1 \times 10^9$ times, we can make the scale of image feature vectors reach 100 million levels. Since the image in MNIST is quite different from the image in Caltech 101, the above operation has no effect on the result of image retrieval. At the same time, we quantize and compress vectors through PQ, OPQ, and LOPQ and then construct the corresponding multi-PQTables.

### 4.2. Experimental Results

In this paper, we mainly confirm the validity and efficiency of the approximate nearest-neighbor search based on the multi-PQTable algorithm by the mean average precision (mAP), the precision rate varying with recall rate, the average retrieval time, and the image retrieval examples. The results and analysis of the experiment are as follows:

1.  Mean Average Precision

We refer to the relevant vector quantization algorithms, i.e., PQ [24], OPQ [25,26], LOPQ [27] and we set a series of multi-PQTable parameters: $K$-means clustering center $K$= 256, size of the compression vector $M$ = 8, number of sub-PQTable $T$ = 2, 4, variable of the generator table $K^*$ = 0, 2, 4. In the multi-PQTable query strategy, we calculate the mAP on the basis of the results of image retrieval. The experimental results are shown in Figure 10. In particular, the original PQ, OPQ, and LOPQ do not have multi-PQTable query strategies.

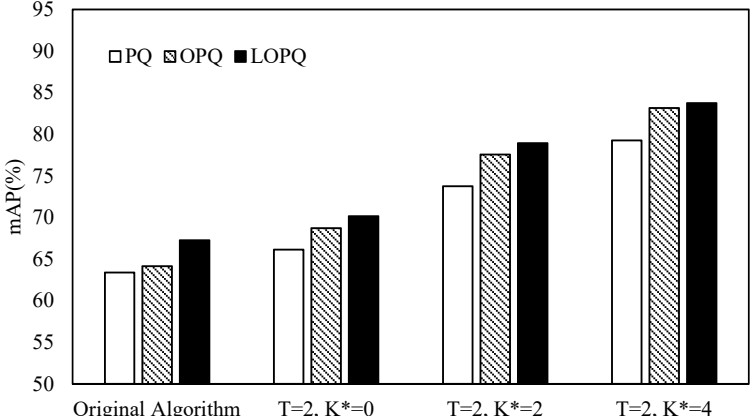

(**a**) The original quantization algorithm and the number of sub-PQTables $T$ = 2. OPQ: Optimized Product Quantization, LOPQ: Locally Optimized Product Quantization.

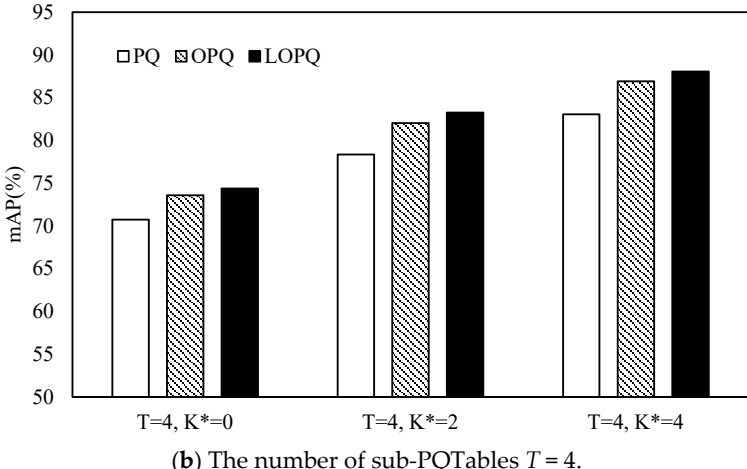

(**b**) The number of sub-PQTables $T$ = 4.

**Figure 10.** The mean average precision (mAP) of the multi-PQTable query strategy varies with the variable $K^*$ of the generator table.

As can be seen from the Figure 10, compared with the original PQ, OPQ, and LOPQ, as the number *T* of the sub-PQTable increases, the mAP of the image retrieval also gradually increases. This shows that the more the sub-PQTables, the lower the index sparseness, the lower the image retrieval failure, the better the image retrieval effect. At the same time, when the variable *K\** of the generator table increases, the set of neighbor vectors generated by the generator table also increases gradually, and the mAP of image retrieval also increases gradually. When the parameters *T* and *K\** are the same, the image retrieval effect of the original vector quantization algorithm is LOPQ > OPQ > PQ, and the image retrieval effect based on the PQTable algorithm is LOPQ > OPQ > PQ. This shows that LOPQ is more advantageous than OPQ and PQ in the approximate nearest-neighbor search.

2.    Precision Rate Varying with the Recall Rate

According to the image retrieval results of PQ, OPQ, and LOPQ and their corresponding multi-PQTable algorithms, we calculate the precision rate varying with the recall rate. The experimental results are shown in Figures 11 and 12.

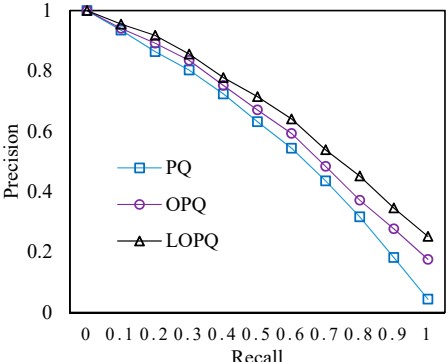

**Figure 11.** The precision of the original quantization algorithms varies with the recall rate.

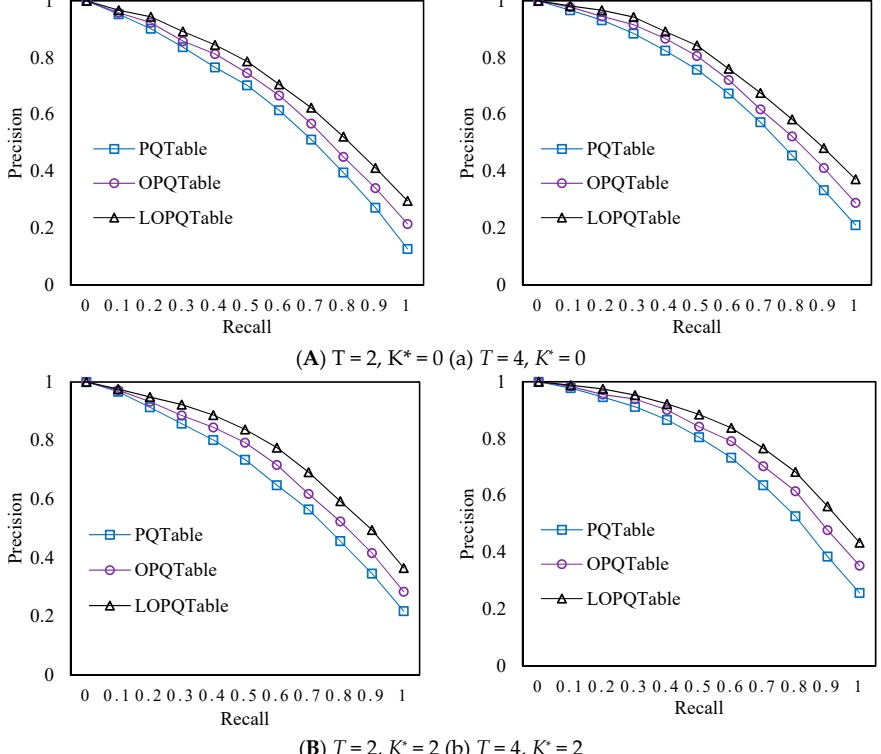

(**A**) T = 2, K\* = 0 (a) *T* = 4, *K\** = 0

(**B**) *T* = 2, *K\** = 2 (b) *T* = 4, *K\** = 2

**Figure 12.** *Cont.*

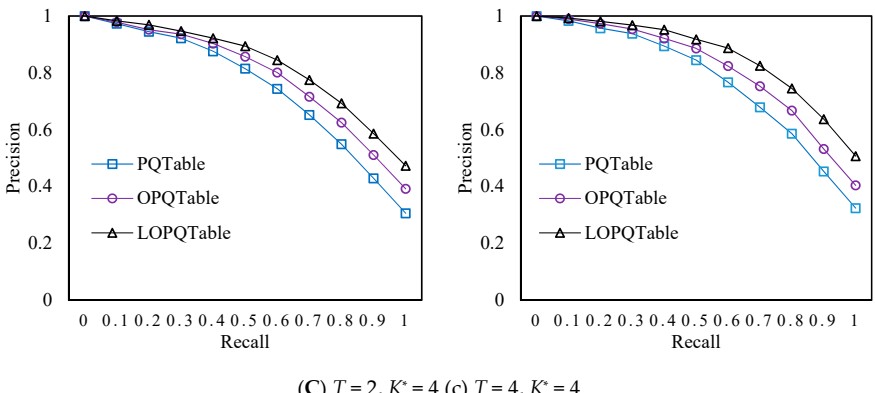

(**C**) $T = 2$, $K^* = 4$ (c) $T = 4$, $K^* = 4$

**Figure 12.** The precision of the multi-PQTable algorithms varies with the recall rate.

From Figure 11, in the original quantization algorithms, PQ decreases relatively faster than OPQ and LOPQ, which indicates that the accuracy of PQ is greatly affected by the recall rate. According to the small graphs (A), (B), (C) or (a), (b), (c) in Figure 12, in the multi-PQTable query strategy, as the generator table parameter K* increases gradually, the set of nearest-neighbor vectors generated by the generator table increases gradually, and the curve decreases slowly, which shows that the accuracy is less and less affected by the recall rate. As can be seen from the small graphs (A) and (a), (B) and (b), (C) and (c), when the recall rate and the parameter $K^*$ are the same, the larger the number $T$ of sub-PQTables is, the harder the sparsity is, and the higher the precision rate of image retrieval is. In addition, when the recall rate and the parameters $K^*$ and $T$ are the same, the precision rate of image retrieval is LOPQ > OPQ > PQ. It also shows that LOPQ has better effects and advantages than OPQ and PQ in approximate nearest-neighbor search.

3. Average Retrieval Time

In the process of image retrieval, we also calculate the average retrieval time. The experimental results are shown in Table 1.

**Table 1.** Mean Average Precision.

| No. | Parameter Settings | Average Retrieval Time ($\times 10^3$/ms) | | |
|---|---|---|---|---|
| | | PQ | OPQ | LOPQ |
| 1 | Original Algorithm | 14.29 | 14.23 | 14.16 |
| 2 | $T = 2$, $K^* = 0$ | 45.93 | 45.86 | 45.77 |
| 3 | $T = 2$, $K^* = 2$ | 63.56 | 62.95 | 62.84 |
| 4 | $T = 2$, $K^* = 4$ | 217.69 | 198.31 | 197.92 |
| 5 | $T = 4$, $K^* = 0$ | 0.004 | 0.002 | 0.003 |
| 6 | $T = 4$, $K^* = 2$ | 0.012 | 0.008 | 0.009 |
| 7 | $T = 4$, $K^* = 4$ | 0.21 | 0.16 | 0.17 |

From the sequence numbers 1, 2, and 5, it can be found that when $T = 2$, the average retrieval time of the multi-PQTable algorithms is greater than that of the original vector quantization algorithm. However, when $T = 4$, the average retrieval time of the multi-PQTable algorithms is much smaller than that of the original vector quantization algorithm, being reduced by about 3500 times. It shows that when $T$ is very small, the multi-PQTable algorithms have no practical effect. Only when $T(T < M)$ is large enough can the retrieval time be reduced and the retrieval efficiency improved. When the parameters $T$ and $K^*$ are the same, there is no significant difference in the average retrieval time between PQ, OPQ, LOPQ and their corresponding multi-PQTable algorithms.

From the sequence numbers 2, 3, and 4 or 5, 6, and 7, as the parameter $K^*$ of the generator table increases, the retrieval time will also increase. The main reason is that when the parameter K* increases,

the size of the nearest-neighbor vector set increases, which leads to the increase of the retrieval time. According to the curves of precision and recall in Figures 11 and 12, the larger the parameter *K\** is, the less the precision rate is affected by the recall rate, and the better the image retrieval effect is, but the longer the average retrieval time is. Therefore, in the actual application process, we need to choose reasonable parameters and balance the retrieval effect and retrieval time, so as to achieve the best performance of the retrieval system.

## 4. Image Retrieval Examples

We selected three categories from the Caltech101 dataset, i.e., airplane, sunflower, and face, and randomly selected an image from these three categories as the query target. We cn obtained the top 10 images from the search results, and the retrieval examples are shown in Figure 13. We extract the image feature vectors through the TSPP-Net model and then construct the index structure of the multi-PQTable correspondingly after quantizing and compressing the vectors. In the approximate nearest-neighbor search, we can quickly and successfully retrieve images with similar content or same categories.

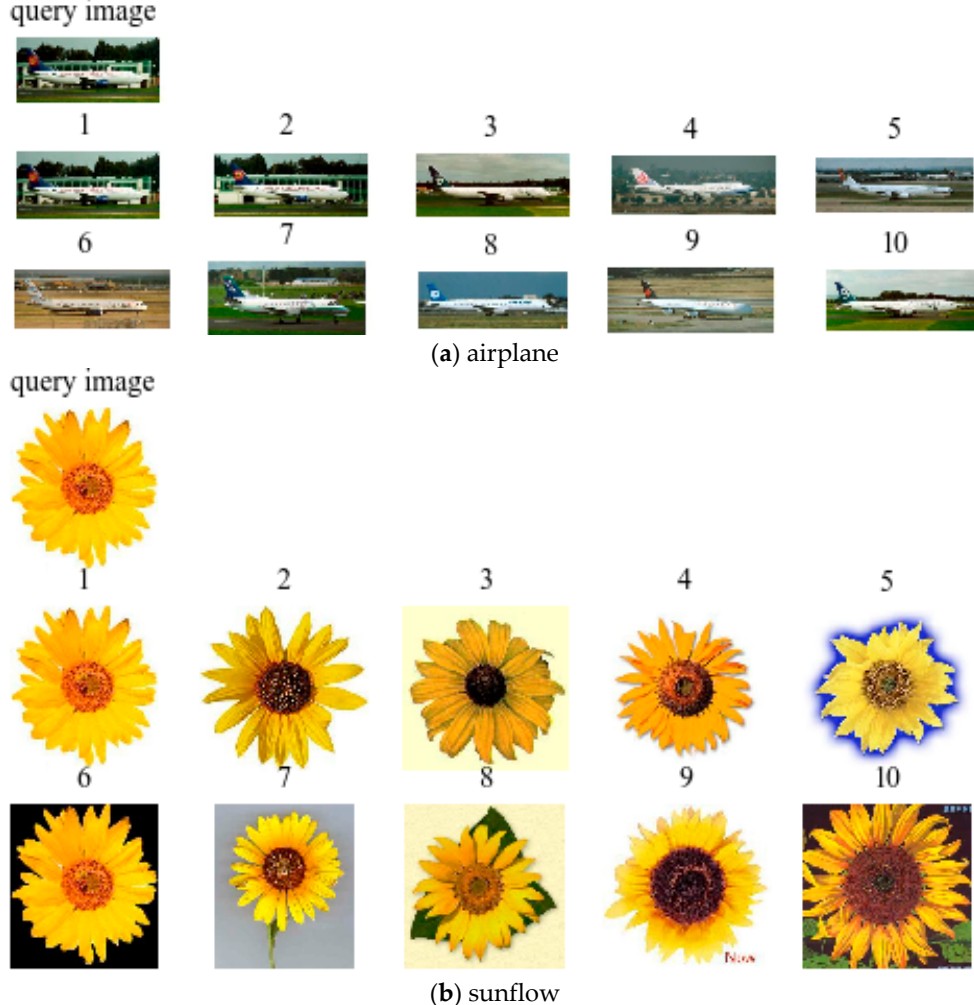

**Figure 13.** *Cont.*

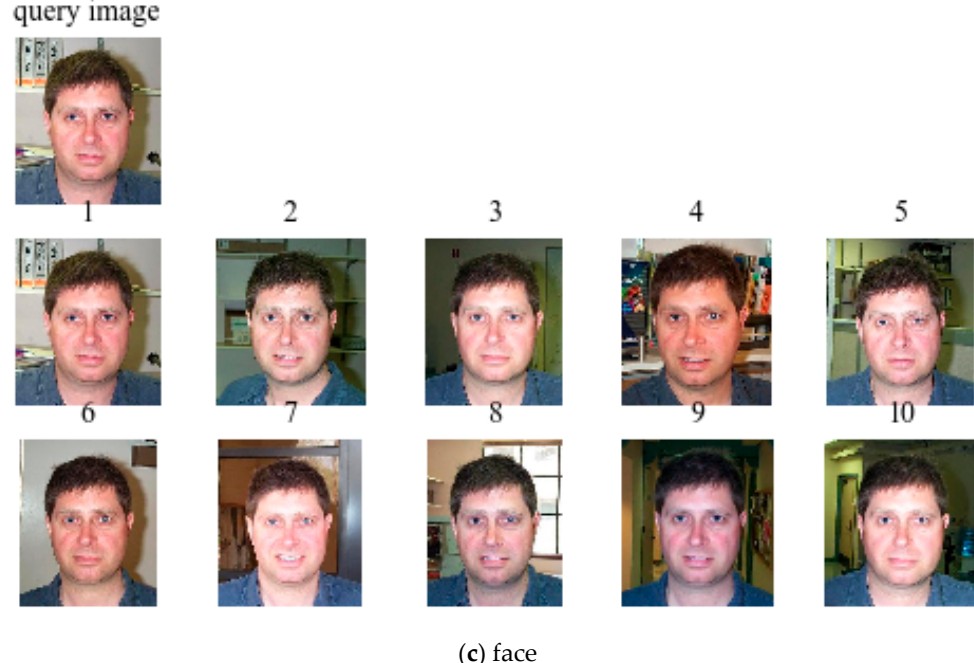

(**c**) face

**Figure 13.** The top 10 images from the search results.

## 5. Conclusions

In this paper, we propose an approximate nearest-neighbor search algorithm based on the PQTable. After quantizing and compressing the image feature vectors by the product quantization algorithm, we can construct the image index structure of the PQTable, which speeds up image retrieval. We also propose a multi-PQTable query strategy for ANN search. Besides, we generate several nearest-neighbor vectors for each sub-compressed vector of the query vector to reduce the failure rate and improve the recall in image retrieval. Through theoretical analysis and experimental verification, it is proved that the multi-PQTable query strategy and the generation of several nearest-neighbor vectors are greatly correct and efficient. Meanwhile, we use the multi-PQTable algorithms for image retrieval and design image similarity detection systems. In the image similarity detection process for the National Natural Science Foundation project, the system can quickly and accurately detect images with similar contents. The system is extremely stable and achieves our desired results.

The vector quantization algorithms, such as PQ, OPQ, LOPQ, can obtain low-dimensional codebooks by quantizing and compressing high-dimensional vectors and achieve good performances in approximate-nearest neighbor search. However, there are some quantization errors in the process of vector quantization and compression via the *K*-means clustering algorithm. Therefore, how to combine deep learning with vector quantization algorithms to learn the distribution of data and obtain a more accurate codebook is a hot topic that needs further research in the future.

**Author Contributions:** Conceptualization, X.Y. and Q.L.; formal analysis, J.L. and Q.L.; funding acquisition, J.L.; investigation, Q.L.; methodology, J.L. and Q.L.; project administration, X.Y. and Q.L.; software, X.Y. and Q.L.; supervision, X.Y.; validation, L.H; visualization, X.Y.; writing (original draft), Q.L. and S.W.; writing (review and editing), Q.L.

**Funding:** This work was supported in part by the National Natural Science Foundation of China (61402165, 61702560), the Key Research Program of Hunan Province (2016JC2018, 2018GK2052), the Natural Science Foundation of Hunan Province (2018JJ2099).

**Conflicts of Interest:** The authors declare no conflict of interest.

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
