# Peer review of "Multi-PQTable for Approximate Nearest-Neighbor Search"

_information, doi:10.3390/info10060190_

Round 1
Reviewer 1 Report
The method is simple and interesting. My understanding of it is that the PQTable is simply a lookup table of all possible compressed vectors, where you can quickly lookup vectors which are exactly the same when compressed with product quantization. The authors correctly identify that this could be an extremely sparse table, and there could be many missed when querying it. I also don't think it's a particularly interesting or novel approach. Lookup tables like this exist, they've just used one in the context of product quantization.
The slightly more interesting proposal is the multi-pq table. I think this is where the paper achieves its novelty. My understanding is that the vector is split into a set of sub vectors, and these are compressed separately. There is then a lookup table for each sub vector, and a potential nearest neighbour only has to share an identical compressed sub vector, which will likely happen more often.
The experiments show a comparison of the method when using different versions of product quantization. I can't see any comparisons with other methods though, so it is hard to know the exact merits of the author's solution. I still think the method description and results are interesting, but maybe some empirical comparison with a state-of-the-art method would be useful. I'm certainly not someone who thinks a paper needs to improve on state-of-the-art to provide a worthwhile and interesting contribution to academic knowledge, so I don't think the author's have to demonstrate their method is better than everybody else's, but I think some sort of experimental comparison is necessary.
The description of the multi-pq table is very hard to follow, and I'm not sure I grasped every detail. Maybe some illustration example of how a vector is compressed would be useful.
Line 137, Q is the feature vector of the query image. This isn't clear.
Overall I think the paper is good enough for publication with some minor changes. The method description could be improved, and some sort of experimental comparison with another method(s) would be useful to the reader.
Author Response
Dear Editors and Reviewers,
We do appreciate your valuable suggestions on our paper entitled “Multi-PQTable for Approximate Nearest Neighbor Search” (Manuscript ID: information- 503401). After receiving the suggestions, we have made great efforts to carefully revise the manuscript according to the comments and suggestions of reviewers, which are rewritten with the red color. The responses to reviewers are as follows:
Minor issues:
Point 1: The slightly more interesting proposal is the multi-pq table. I think this is where the paper achieves its novelty. My understanding is that the vector is split into a set of sub vectors, and these are compressed separately. There is then a lookup table for each sub vector, and a potential nearest neighbour only has to share an identical compressed sub vector, which will likely happen more often.
The experiments show a comparison of the method when using different versions of product quantization. I can't see any comparisons with other methods though, so it is hard to know the exact merits of the author's solution. I still think the method description and results are interesting, but maybe some empirical comparison with a state-of-the-art method would be useful. I'm certainly not someone who thinks a paper needs to improve on state-of-the-art to provide a worthwhile and interesting contribution to academic knowledge, so I don't think the author's have to demonstrate their method is better than everybody else's, but I think some sort of experimental comparison is necessary.
Response 1: Thank you for your suggestions. This paper proposes a multi-PQTable algorithm on the basis of the PQ algorithm. Therefore, our experiment verifies the correctness of the algorithm by PQ, OPQ and LOPQ. In the experimental process, we also use multi-PQTable to compare with the original vector quantization algorithm, including PQ, OPQ and LOPQ.
Point 2: The description of the multi-pq table is very hard to follow, and I'm not sure I grasped every detail. Maybe some illustration example of how a vector is compressed would be useful.
Response 2: Thank you for your suggestions. In the Section 2, we introduce the process of vector quantization through PQ algorithm in detail. The detailed process can be referred to in the following references:
[24] Jégou H, Douze M, Schmid C. Product Quantization for Nearest Neighbor Search[J]. IEEE Transactions on Pattern Analysis & Machine Intelligence, 2011, 33(1):117-128.
[25] Ge T, He K, Ke Q, et al. Optimized Product Quantization for Approximate Nearest Neighbor Search[C]. IEEE Conference on Computer Vision and Pattern Recognition. IEEE Computer Society, 2013:2946-2953.
[26] Ge T, He K, Ke Q, et al. Optimized Product Quantization[J]. IEEE Transactions on Pattern Analysis & Machine Intelligence, 2014, 36(4):744-755.
[27] Kalantidis Y, Avrithis Y. Locally Optimized Product Quantization for Approximate Nearest Neighbor Search[C]. Computer Vision & Pattern Recognition. 2014.:2329-2336.
Point 3: Line 137, Q is the feature vector of the query image. This isn't clear.
Response 3: Thank you for your suggestions. We also added a sentence in the Section 3.1 and the contents are as follows:
where Q is the feature vector of query image Iq, Xi=[xi1, xi2, ... , xiD] and D is the dimension of the feature vector.
Point 4: Overall I think the paper is good enough for publication with some minor changes. The method description could be improved, and some sort of experimental comparison with another method(s) would be useful to the reader.
Response 4: Thank you for your suggestions. We have made great efforts to carefully revise the manuscript according to the comments and suggestions, which are rewritten with the red color.
Thank you for your suggestions. We have revised this problem in the revised paper. We hope that the revised paper can satisfy the requirements of the journal.
Sincerely yours,
Authors: Xinpan Yuan, Qunfeng Liu, Jun Long *, Lei Hu, Songlin Wang
Reviewer 2 Report
This paper proposes a Multi-PQTable based approximate nearest neighbor search approach for speeding up image retrieval process. Detailed previous work the proposed approach based are described, and experiment results on data set are carried out for evaluating the accuracy and efficiency of the approach. The paper reads well.
A few issues/suggestions:
Only a few categories from small dataset are tested in the experimental stage. What are the approach's performance for the other categories or other larger datasets? Suggest the authors to try more experiments.
In Section 4.2.1, there are several variables such as K, M, T. What are the criteria of assigning values to these variables, or why the current values are preferred instead of others? Suggest the authors to explain a little bit more.
A few suggestions on the write up:
The caption of Fig.11(a) is on a different page with the figure. Suggest to adjust the figure position.
In Line 70, Page 2, the third point shall not be listed as a contribution. Theoretical analysis and experiments are basic component of each paper.
Suggest to add a few references on how to parallelize image retrieving process: (1) Hu, F., Zhu, Z., Mejia, J., Tang, H., & Zhang, J. (2017). "Real-time indoor assistive localization with mobile omnidirectional vision and cloud GPU acceleration", AIMS Electronics and Electrical Engineering, Nov, 2017. (2)Zhou Bing; Yang Xin-xin, "A content-based parallel image retrieval system,"International Conference on Computer Design and Applications (ICCDA), 2010, vol.1, no., pp.V1-332,V1-336, 25-27 June 2010.
Suggest to double check the reference formats, e.g. some titles have all words capitalized, such as 1, 2, 3, while many others only have the first word capitalized, such as 6, 7, 10, 11.
Author Response
Dear Editors and Reviewers,
We do appreciate your valuable suggestions on our paper entitled “Multi-PQTable for Approximate Nearest Neighbor Search” (Manuscript ID: information- 503401). After receiving the suggestions, we have made great efforts to carefully revise the manuscript according to the comments and suggestions of reviewers, which are rewritten with the red color. The responses to reviewers are as follows:
Minor issues:
Point 1: In Section 4.2.1, there are several variables such as K, M, T. What are the criteria of assigning values to these variables, or why the current values are preferred instead of others? Suggest the authors to explain a little bit more.
Response 1: Thank you for your suggestions. We also added a sentence in the section 4.2 and the contents are as follows:
We refer to the relevant vector quantization algorithms, such as PQ [24], OPQ [25] [26],LOPQ [27] and we set a series of multi-PQTable parameters.
Point 2: The caption of Fig.11(a) is on a different page with the figure. Suggest to adjust the figure position.
Response 2: Thank you for your suggestions. We have adjusted the figure position.
Point 3: In Line 70, Page 2, the third point shall not be listed as a contribution. Theoretical analysis and experiments are basic component of each paper.
Response 3: Thank you for your suggestions. We have removed the third point in line 70.
Point 4: Suggest to add a few references on how to parallelize image retrieving process: (1) Hu, F., Zhu, Z., Mejia, J., Tang, H., & Zhang, J. (2017). "Real-time indoor assistive localization with mobile omnidirectional vision and cloud GPU acceleration", AIMS Electronics and Electrical Engineering, Nov, 2017. (2)Zhou Bing; Yang Xin-xin, "A content-based parallel image retrieval system,"International Conference on Computer Design and Applications (ICCDA), 2010, vol.1, no., pp.V1-332,V1-336, 25-27 June 2010.
Response 4: Thank you for your suggestions. We also added a sentence in the section 1 and the contents are as follows:
Besides, there's also some research work about how to parallelize image retrieving process [32][33].
[32] Hu, F., Zhu, Z., Mejia, J., Tang, H., Zhang, J. Real-time indoor assistive localization with mobile omnidirectional vision and cloud GPU acceleration[J], AIMS Electronics and Electrical Engineering, 2017.
[33] Zhou Bing, Yang Xin-xin. A content-based parallel image retrieval system[C]. International Conference on Computer Design and Applications (ICCDA), 2010(1):332 -336.
Point 5: Suggest to double check the reference formats, e.g. some titles have all words capitalized, such as 1, 2, 3, while many others only have the first word capitalized, such as 6, 7, 10, 11.
Response 5: Thank you for your advices. We have unified the reference formats. Some titles have all words capitalized.
Thank you for your suggestions. We have revised this problem in the revised paper. We hope that the revised paper can satisfy the requirements of the journal.
Sincerely yours,
Authors: Xinpan Yuan, Qunfeng Liu, Jun Long *, Lei Hu, Songlin Wang
Round 2
Reviewer 2 Report
The proposed issues are resolved by the authors.